# Towards Feature Space Adversarial Attack

## Abstract

We propose a new type of adversarial attack to Deep Neural Networks (DNNs) for image classification. Different from most existing attacks that directly perturb input pixels. Our attack focuses on perturbing abstract features, more specifically, features that denote styles, including interpretable styles such as vivid colors and sharp outlines, and uninterpretable ones. It induces model misclassfication by injecting style changes insensitive for humans, through an optimization procedure. We show that state-of-the-art pixel space adversarial attack detection and defense techniques are ineffective in guarding against feature space attacks.

## 1 Introduction

Adversarial attacks are a prominent threat to the broad application of Deep Neural Networks (DNNs). In the context of classification applications, given a pre-trained model $M$ and a benign input $x$ of some output label $y$, adversarial attack perturbs $x$ such that $M$ misclassifies the perturbed $x$. The perturbed input is called *adversarial example*. Such perturbations are usually bounded by some distance norm such that they are not perceptible by humans. Ever since it was proposed in (Szegedy et al., 2014), there has been a large body of research that develops various methods to construct adversarial examples, e.g., (Carlini & Wagner, 2017; Madry et al., 2018), with different modalities such as images (Carlini & Wagner, 2017), audio (Qin et al., 2019), text (Ebrahimi et al., 2018), and video (Li et al., 2019), detect adversarial examples (Tao et al., 2018; Ma et al., 2019), and use adversarial examples to harden the models (Madry et al., 2018; Zhang et al., 2019).

However, most existing attacks are in the pixel space, that is, the perturbations occur directly in the pixel space, and the pixel level differences between the adversarial example and the original input are bounded. In this paper, we illustrate that adversarial attack can be conducted in the feature space. The underlying assumption (in the context of image classification) is that during training, a DNN may extract a large number of abstract features. While many of them denote critical characteristics of the object, some of them are secondary, for example, the different styles of an image (e.g., vivid colors versus pale colors, sharp outlines versus blur outlines). These secondary features may play an improperly important role in model prediction. As a result, feature space attack can inject such secondary features, which are not simple pixel perturbation, but rather functions over the given benign input, to induce model misclassification. Since humans are not sensitive to these features, the resulted adversarial examples look very natural from humans' perspective. As many of these features are pervasive, the resulted pixel space perturbation may be relatively much more substantial than existing pixel space attacks. As such, pixel space detection and hardening techniques are ineffective for feature space attacks (see §4). Figure 1 shows a number of adversarial examples generated by our technique, their comparison with the original examples, and the pixel space distances. Observe that while the distances are much larger compared to those in pixel space attacks, the adversarial examples are completely natural, or even indistinguishable from the original inputs in humans' eyes. The contrast of the benign-adversarial pairs illustrates that the malicious perturbations largely co-locate with the primary content features, denoting imperceptible tweaking of these features.

Under the hood, we consider that the activations of an inner layer represent a set of abstract features, including those primary and secondary. To avoid generating adversarial examples that are unnatural, we refrain from tampering with the primary features (or *content features*) and focus on perturbing the secondary *style features*. Inspired by the recent advance in style transfer (Huang & Belongie, 2017), the *mean* and *variance* of activations are considered the style. As such, we focus on perturbing the means and variances while preserving the *shape* of the activation values (i.e., the up-and-downs of these values and the relative scale of such up-and-downs). We use gradient driven optimization to search for the style perturbations that can induce misclassification. Since our threat model is the same as existing pixel space attacks, that is, the attack is launched by providing the adversarial example to the model. An important step is to translate the activations with style changes back to a naturally looking pixel space example. We address the problem by considering

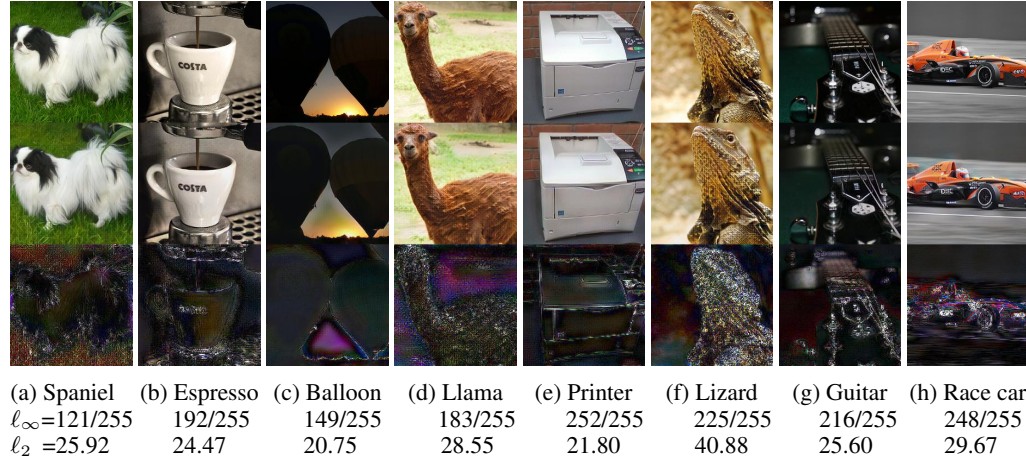

| (a) Spaniel | (b) Espresso | (c) Balloon | (d) Llama | (e) Printer | (f) Lizard | (g) Guitar | (h) Race car |
|---|---|---|---|---|---|---|---|
| $\ell_\infty$=121/255 | 192/255 | 149/255 | 183/255 | 252/255 | 225/255 | 216/255 | 248/255 |
| $\ell_2$ =25.92 | 24.47 | 20.75 | 28.55 | 21.80 | 40.88 | 25.60 | 29.67 |

Figure 1: Examples by feature space attack. The top row presents original images. The middle row denotes adversarial samples. The third row shows the pixel-wise difference ($\times 3$) between original images and adversarial samples. The $\ell_\infty$ and $\ell_2$ norms are shown on the bottom.

the differences of any pair of training inputs of the same class as the possible style differences, and pre-training a decoder that can automatically impose styles in the pixel space based on the style feature perturbation happening in an inner layer. We propose two concrete feature space attacks, one to enhance styles and the other to impose styles constituted from a set of pre-defined style prototypes.

We evaluate our attacks on 3 datasets and 7 models. We show that feature space attacks can effectively generate adversarial samples that evade 5 state-of-the-art detection/defense approaches. Particularly, our proposed attack can reduce the detection rate of a state-of-the-art approach (Roth et al., 2019) to 0.04% on the CIFAR-10 dataset, and the prediction accuracy of a model hardened by a state-of-art adversarial training technique (Xie et al., 2019) to 1.25% on ImageNet. Moreover, we observe that despite the large distance introduced in the pixel space, the distances in feature space are similar or even smaller than those in $\ell$-norm based attacks. The generated adversarial examples have only natural, and in many cases, human imperceptible style differences compared with the original inputs, demonstrating the practicality of the attacks.

This work only demonstrates the feasibility of feature space attacks. The features we are attacking are relatively simple. In the future, we expect more research on complicated feature space attacks.

## 2 BACKGROUND AND RELATED WORK

**Style Transfer.** Huang & Belongie (2017) proposed to transfer the style from a (source) image to another (target) that may have different content such that the content of the target image largely retains while features that are not essential to the content align with those of the source image. Specifically, given an input image, say the portrait of actor Brad Pitt, and a style picture, e.g., a drawing of painter Vincent van Gogh, the goal of style transfer is to produce a portrait of Brad Pitt that looks like a picture painted by Vincent van Gogh. Existing approaches leverage various techniques to achieve this purpose. Gatys et al. (2016) utilized the feature representations in convolutional layers of a DNN to extract *content features* and *style features* of input images. Given a random white noise image, the algorithm feeds the image to the DNN to obtain the corresponding content and style features. The content features from the white noise image are compared with those from a content image, and the style features are contrasted with those from a style image. It then minimizes the above two differences to transform the noise image to a content image with style. Due to the inefficiency of this optimization process, researchers replace it with a neural network that is trained to minimize the same objective (Li & Wand, 2016; Johnson et al., 2016). Further study extends these approaches to synthesize more than just one fixed style (Dumoulin et al., 2017; Li et al., 2017). Huang & Belongie (2017) introduced a simple yet effective approach, which can efficiently enable arbitrary style transfer. This approach proposes an *adaptive instance normalization* (AdaIN) layer that aligns the mean and variance of the content features with those of the style features.

**Adversarial Attacks.** In the context of image classification, given an input $\boldsymbol{x}$, and a DNN model $M(\cdot)$, an adversary produces a sample $\boldsymbol{x}'$ such that $M(\boldsymbol{x}') \neq M(\boldsymbol{x})$. If the output label $y$ is chosen by the adversary in advance, then it is called a *targeted* attack, i.e., $M(\boldsymbol{x}')=y \neq M(\boldsymbol{x})$; Otherwise,

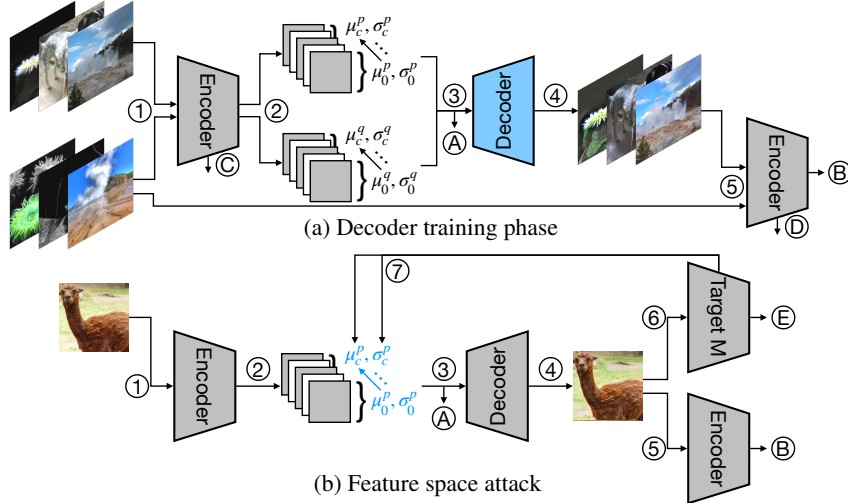

Figure 2: Procedure of feature space adversarial attack. Two phases are involved during the attack generation process: (a) decoder training phase and (b) feature space attack phase.

it is an *untargeted* attack. Existing attacks, targeted or untargeted, adopt $\ell$-norm based metrics to measure the magnitude of introduced perturbation (small values correspond to little perceptibility to humans). Three $\ell$-norm metrics are commonly used: $\ell_0$, $\ell_2$, and $\ell_\infty$. Specifically, $\ell_0$ metric gauges the number of pixels modified when introducing the perturbation. Let $\boldsymbol{\delta} = \boldsymbol{x}' - \boldsymbol{x}$ be the perturbation, $\ell_0$ can be calculated using $\|\boldsymbol{\delta}\|_0 = \left| \{i | \delta_i \neq 0\} \right|$. The $\ell_2$ metric is the Euclidean distance which measures the aggregated pixel changes. It is computed with the following equation: $\|\boldsymbol{\delta}\|_2 = \sqrt{\sum_i \delta_i^2}$. The $\ell_\infty$ metric gauges the maximum change among all the modified pixels. The measurement of $\ell_\infty$ is processed with $\|\boldsymbol{\delta}\|_\infty = \max_i |\delta_i|$. All three metrics are widely used as the standard measurement for evaluating stealthiness of adversarial attacks (Carlini & Wagner, 2017; Madry et al., 2018) and robustness of DNN models (Xu et al., 2018; Ma et al., 2019). In this paper, however, we find that these traditional *pixel-space* $\ell$-norm metrics do not fully reflect the stealthiness of feature space attacks. Figure 1 showcases adversarial samples generated using our feature space attack. All the generated samples have very large $\ell_\infty$ bound ($>120/255$), which is at least 7.5 times larger than the commonly used bounds in the literature of adversarial attacks (Kurakin et al., 2017; Roth et al., 2019) ($<16/255$ on ImageNet dataset). The $\ell_2$ bound is more than 20, substantially larger than those in the literature as well, i.e., less than 0.5 in (Rony et al., 2019; Lecuyer et al., 2019; Cohen et al., 2019). However, observe that the adversarial samples have very natural style differences compared to the original inputs.

## 3 FEATURE SPACE ATTACK

**Overview.** We aim to demonstrate that perturbation in the feature space can lead to model misbehavior, which existing pixel space hardening techniques cannot effectively defend against. The hypothesis is that during training, the model picks up numerous features, many of which do not describe the key characteristics (or *content*) of the object, but rather human imperceptible features such as styles. These subtle features may play an improperly important role in model prediction. As a result, injecting such features to a benign image can lead to misclassification. However, the feature space is not exposed to attackers such that they cannot directly perturb features. Therefore, a prominent challenge is to derive the corresponding pixel space mutation that appears natural to humans while leading to the intended feature space perturbation, and eventually the misclassification. In particular, the attack comprises two phases: (1) training a decoder that can translate feature space perturbation to pixel level changes that look natural for humans; (2) launching the attack by first using gradient based optimization to identify feature space perturbation that can cause misclassification and then using the decoder to generate the corresponding adversarial example. Inspired by style transfer techniques, we consider a much confined feature perturbation space – *style perturbation*. Specifically, as in (Huang & Belongie, 2017), we consider the *mean* and *variance* of the activations of an inner layer denote the style of the features in that layer whereas the activations themselves denote the content features. We hence perturb the mean and variance of content features by performing a predefined transformation that largely preserves the shape of the features while changing

the mean and variance. The decoder then decodes the perturbed feature values to an image closely resembles the original image with only style differences that appear natural to humans but causing model misclassification.

Figure 2 illustrates the workflow of the proposed attack. In the Decoder training phase (a), a set of image pairs with each pair from the same class (and hence their differences can be intuitively considered as style differences) are fed to a fixed Encoder that essentially consists of the first a few layers of a pre-trained model (e.g., VGG-19) (step ①). The Encoder produces the internal embeddings of the two respectively images, which correspond to the activation values of some inner layer in the pre-trained model, e.g., conv4_1 (step ②). Each internal embedding consists of a number of matrices, one for every channel. For each embedding matrix, the mean and variance are computed. We use these values from the two input images to produce the integrated embedding Ⓐ (step ③), which will be discussed in details later in this section. Intuitively, it is generated by performing a shape-preserving transformation of the upper matrix so that it retains the content features denoted by the upper matrix while having the mean and variance of the lower matrix (i.e., the style denoted by the lower matrix). We employ a Decoder to reconstruct a raw image from Ⓐ at step ④, which is supposed to have the content of the upper image (called the *content image*) and the style of the lower image (called the *style image*). To enable good reconstruction performance, two losses are utilized for optimizing the Decoder. The first one is the *content loss*. Specifically, at step ⑤ the reconstructed image is passed to the Encoder to acquire the reconstructed embedding Ⓑ, and then the difference between the integrated embedding Ⓐ and the reconstructed embedding Ⓑ is minimized. The second one is the *style loss*. Particularly, the means and variances of a few selected internal layers of the Encoder are computed for both the generated image and the original style image. The difference of these values of the two images is minimized. The Decoder optimization process is conducted on the original training dataset of target model $M$ (under attack). Intuitively, the decoder is trained to understand the style differences so that it can decode feature style differences to realistic pixel space style differences, by observing the possible style differences.

When launching the attack ((b) in Figure 2), a test input image is fed to the Encoder and goes through the same process as in the Decoder training phase. The key differences are that only one input image is required and the Decoder is fixed in this phase. Given a target model $M$ (under attack), the reconstructed image is fed to $M$ at step ⑥ to yield prediction Ⓔ. As the attack goal is to induce $M$ to misclassify, the difference between prediction Ⓔ and a target output label (different from Ⓔ) is considered the *adversarial loss* for launching the attack. In addition, the content loss between Ⓐ and Ⓑ is also included. The attack updates the means and variances of embedding matrices at step ⑦ with respect to the adversarial loss and style loss. The final reconstructed image that induces the target model $M$ to misclassify is a successful adversarial sample.

## 3.1 DEFINITIONS

In this section, we formally define feature space attack. Considering a typical classification problem, where the samples $x \in \mathbb{R}^d$ and corresponding label $y \in \{0, 1, \ldots, n\}$ jointly obey a distribution $\mathcal{D}(x, y)$. Given a classifier $M : \mathbb{R}^d \to \{0, 1, \ldots, n\}$ with parameter $\theta$. The goal of training is to find the best parameter $\arg\max_\theta P_{(x,y)\sim\mathcal{D}}[M(x; \theta) = y]$. Empirically, people associate a continuous loss function $\mathcal{L}_{M,\theta}(x, y)$, e.g. cross-entropy, to measure the difference between the prediction and the true label. And the goal is rewritten as $\arg\min_\theta \mathbb{E}_{(x,y)\sim\mathcal{D}}[\mathcal{L}_{M,\theta}(x, y)]$. We use $\mathcal{L}_M$ in short for $\mathcal{L}_{M,\theta}$ in the following discussion. In adversarial learning, the adversary can introduce a perturbation $\delta \in \mathbb{S} \subset \mathbb{R}^d$ into the natural samples $(x, y) \sim \mathcal{D}$. For a given sample $x$ with label $y$, an adversary chooses the most malicious perturbation $\arg\max_{\delta\in\mathbb{S}} \mathcal{L}_M(x + \delta, y)$ to make the classifier $M$ predict incorrectly. Normally $\mathbb{S}$ is confined as an $\ell_p$-ball centered on $0$. In this case, the $\ell_p$ norm of pixel space differences measures the distance between adversarial samples (i.e., $x + \delta$ that causes misclassification) and the original samples. Thus we refer to this attack model as the *pixel space attack*. Most existing adversarial attacks fall into this category. Different from adding bounded perturbation in the pixel space, feature space attack applies perturbation in the feature space such that an encoder (to extract the feature representation of the benign input) and a decoder function (that translates perturbed feature values to a naturally looking image that closely resembles the original input in humans' perspective).

Formally, consider an encoder function $f : \mathbb{R}^d \to \mathbb{R}^e$ and a decoder function $f^{-1} : \mathbb{R}^e \to \mathbb{R}^d$. The former encodes a sample to an embedding $b \in \mathbb{R}^e$ and the latter restores an embedding back to a

sample. A perturbation function $a \in \mathbb{A} : \mathbb{R}^e \to \mathbb{R}^e$ transforms a given embedding to another. For a given sample $\boldsymbol{x}$, the adversary chooses the best perturbation function to make the model $M$ predict incorrectly.

$$\max_{a \in \mathbb{A}} \mathcal{L}_M[f^{-1} \circ a \circ f(\boldsymbol{x}), y]. \tag{1}$$

Functions $f$ and $f^{-1}$ need to satisfy additional properties to ensure the attack is meaningful. We call them the *wellness properties* of encoder and decoder.

*Wellness of Encoder $f$.* In order to get a meaningful embedding, there ought to exist a well-functioning classifier $g$ based on the embedding, with a prediction error rate less than $\delta_1$.

$$\exists g : \mathbb{R}^e \to \{0, 1, \ldots, n\}, P_{(\boldsymbol{x}, y) \sim \mathcal{D}}[g(f(\boldsymbol{x})) = y] \geq 1 - \delta_1, \text{ for a given } \delta_1. \tag{2}$$

In practice, this property can be easily satisfied as one can construct $g$ from a well-functioning classifier $M$, by decomposing $M = M_2 \circ M_1$ and take $M_1$ as $f$ and $M_2$ as $g$.

*Wellness of Decoder $f^{-1}$.* Function $f^{-1}$ is essentially a translator that translates what the adversary has done on the embedding back to a sample in $\mathbb{R}^d$. We hence require that for all possible adversarial transformation $a \in \mathbb{A}$, $f^{-1}$ ought to retain what the adversary has applied to the embedding in the restored sample.

$$\forall a \in \mathbb{A}, \text{ let } B^a = a \circ f(\boldsymbol{x}), E_{(\boldsymbol{x}, y) \sim \mathcal{D}} ||f \circ f^{-1}(B^a) - B^a||_2 \leq \delta_2, \text{ for a given } \delta_2. \tag{3}$$

This ensures a decoded (adversarial) sample induce the intended perturbation in the feature space. Note that $f^{-1}$ can always restore a benign sample back to itself. This is equivalent to requiring the identity function in the perturbation function set $\mathbb{A}$.

Given $(f, f^{-1}, \mathbb{A})$ satisfying the aforementioned properties, we define Equation 1 as a feature space attack. Under this definition, pixel space attack is a special case of feature space attack. For an $\ell_p$-norm $\epsilon$-bounded pixel space attack, i.e., $\mathbb{S} = \{||\boldsymbol{\delta}||_p \leq \epsilon\}$, we can rewrite it as a feature-space attack. Let encoder $f$ and decoder $f^{-1}$ be an identity function and let $\mathbb{A} = \cup_{||\boldsymbol{\delta}||_p \leq \epsilon} \{a : a(\boldsymbol{m}) = \boldsymbol{m} + \boldsymbol{\delta}\}$.

$$\begin{aligned} \text{pixel space attack} &\overset{\text{def}}{=} \max_{||\boldsymbol{\delta}||_p \leq \epsilon} \mathcal{L}_M(\boldsymbol{x} + \boldsymbol{\delta}, y) = \max_{a \in \mathbb{A}} \mathcal{L}_M[a(\boldsymbol{x}), y] \\ &= \max_{a \in \mathbb{A}} \mathcal{L}_M[f^{-1} \circ a \circ f(\boldsymbol{x}), y] \overset{\text{def}}{=} \text{Equation 1}. \end{aligned} \tag{4}$$

One can easily verify the wellness of $f$ and $f^{-1}$. Note that the stealthiness of feature space attack depends on the selection of $\mathbb{A}$, analogous to that the stealthiness of pixel space attack depending on the $\ell_p$ norm. Next, we demonstrate two stealthy feature space attacks.

## 3.2 ATTACK DESIGN

### 3.2.1 DECODER TRAINING

Our decoder design is illustrated in Figure 2a. It is inspired by style transfer in (Huang & Belongie, 2017). To train the decoder, we enumerate all the possible pairs of images in each class in the original training set and use these pairs as a new training set. We consider each pair has the same content features (as they belong to the same class) and hence their differences essentially denote style differences. By training the decoder on all possible style differences (in the training set) regardless the output classes, we have a general decoder that can recognize and translate arbitrary style perturbation. Formally, given a normal image $\boldsymbol{x}_p$ and another image $\boldsymbol{x}_q$ from the same class as $\boldsymbol{x}_p$, the training process first passes them through a pre-trained Encoder $f$ (e.g., VGG-19) to obtain embeddings $B^p = f(\boldsymbol{x}_p), B^q = f(\boldsymbol{x}_q) \in \mathbb{R}^{H \cdot W \cdot C}$, where $C$ is the channel size, and $H$ and $W$ are the height and width of each channel. For each channel $c$, the mean and variance are computed across the spatial dimensions (step ② in Figure 2a). That is,

$$\mu_{B_c} = \frac{1}{HW} \sum_{h=1}^{H} \sum_{w=1}^{W} B_{hwc}, \qquad \sigma_{B_c} = \sqrt{\frac{1}{HW} \sum_{h=1}^{H} \sum_{w=1}^{W} (B_{hwc} - \mu_{B_c})^2}. \tag{5}$$

We combine the embeddings $B^p$, $B^q$ from the two input images using the following equation:

$$\forall c \in [1, 2, ..., C], B_c^o = \sigma_{B_c^q} \left( \frac{B_c^p - \mu_{B_c^p}}{\sigma_{B_c^p}} \right) + \mu_{B_c^q}, \tag{6}$$

where $B_c^o$ is the result embedding of channel $c$. Intuitively, the transformation retains the shape of $B^p$ while enforcing the mean and variance of $B^q$. $B^o$ is then fed to the Decoder $f^{-1}$ for reconstructing the image with the content of $\boldsymbol{x}_p$ and the style of $\boldsymbol{x}_q$ (steps ③ & ④ in Figure 2a). In order to generate a realistic image, the reconstructed image is passed to Encoder $f$ to acquire the reconstructed embedding $B^r = f \circ f^{-1}(B^o)$ (step ⑤). The difference between the combined embedding $B^o$ and the reconstructed embedding $B^r$, called the *content loss*, is minimized using the following equation during the Decoder training:

$$\mathcal{L}_{\text{content}} = ||B^r - B^o||_2. \tag{7}$$

In addition, some internal layers of Encoder $f$ are selected, whose means and variances (computed by Equation 5) are used for representing the style of input images. The difference of these values between the style image $\boldsymbol{x}_q$ and the reconstructed image $\boldsymbol{x}_r$, called the *style loss*, is minimized when training the Decoder. It is defined as follows:

$$\mathcal{L}_{\text{style}} = \sum_{i \in L} ||\mu(\phi_i(\boldsymbol{x}_q)) - \mu(\phi_i(\boldsymbol{x}_r))||_2 + \sum_{i \in L} ||\sigma(\phi_i(\boldsymbol{x}_q)) - \sigma(\phi_i(\boldsymbol{x}_r))||_2, \tag{8}$$

where $\phi_i(\cdot)$ denotes layer $i$ of Encoder $f$ and $L$ the set of layers considered. In this paper, $L$ consists of conv1_1, conv2_1, conv3_1 and conv4_1 for the ImageNet dataset, and conv1_1 and conv2_1 for the CIFAR-10 and SVHN datasets. $\mu(\cdot)$ and $\sigma(\cdot)$ denote the mean and the variance, respectively. The Decoder training is to minimize $\mathcal{L}_{\text{content}} + \mathcal{L}_{\text{style}}$.

### 3.2.2 TWO FEATURE SPACE ATTACKS

Recall in the attack phase (Figure 2b), the encoder and decoder are fixed. The style features of a benign image are perturbed while the content features are retained, aiming to trigger misclassification. The pre-trained decoder then translates the perturbed embedding back to an adversarial sample. During perturbation, we focus on minimizing two loss functions. The first one is the adversarial loss $\mathcal{L}_M$ whose goal is to induce misclassification. The second one is similar to the content loss in the Decoder training (Equation 7). Intuitively, although the decoder is trained in a way that it is supposed to decode with minimal loss, arbitrary style perturbation may still cause substantial loss. Hence, such loss has to be considered and minimized during style perturbation.

With two different sets of transformations $\mathbb{A}$, we devise two respective kinds of feature space attacks, *feature augmentation attack* and *feature interpolation attack*. For feature augmentation attack, attacker can change both the mean and standard deviation of each channel of the benign embedding independently. The boundary of increments or decrements are set by $\ell_\infty$-norm under logarithm scale (to achieve stealthiness). Specifically, given two perturbation vectors $\boldsymbol{\tau}^\mu$ for the mean and $\boldsymbol{\tau}^\sigma$ for the variance, both have the same dimension $C$ as the embedding (denoting the $C$ channels) and are bounded by $\epsilon$, the list of possible transformations $\mathbb{A}$ is defined as follows.

$$\mathbb{A} = \cup_{||\boldsymbol{\tau}^\sigma||_\infty \le \epsilon \text{ and } ||\boldsymbol{\tau}^\mu||_\infty \le \epsilon, \, \boldsymbol{\tau}^\sigma \text{ and } \boldsymbol{\tau}^\mu \in \mathbb{R}^C} \left\{ a : a(B)_{h,w,c} = e^{\boldsymbol{\tau}_c^\sigma}(B_{h,w,c} - \mu_{B_c}) + e^{\boldsymbol{\tau}_c^\mu} \mu_{B_c} \right\} \tag{9}$$

Note that $\mu_B$ denotes the means of embedding $B$ for the $C$ channels. The subscript $c$ denotes a specific channel. The transformation essentially enlarges the variance of the embedding at channel $c$ by a factor of $e^{\boldsymbol{\tau}_c^\sigma}$ and the mean by a factor of $e^{\boldsymbol{\tau}_c^\mu}$.

For the feature interpolation attack, the attacker provides $k$ images as the style feature prototypes. Let $\mathbb{S}_\mu, \mathbb{S}_\sigma$ be the simplex determined by $\cup_{i \in [1,2,...,k]} \mu_{f(\boldsymbol{x}_i)}$ and $\cup_{i \in [1,2,...,k]} \sigma_{f(\boldsymbol{x}_i)}$ respectively. The attacker can modify the vectors of $\mu_B$ and $\sigma_B$ to be any point on the simplex.

$$\mathbb{A} = \cup_{\sigma_i \in \mathbb{S}_\sigma, \mu_i \in \mathbb{S}_\mu} \left\{ a : a(B)_{h,w,c} = \sigma_i \cdot \frac{B_{h,w,c} - \mu_{B_c}}{\sigma_{B_c}} + \mu_i \right\} \tag{10}$$

Intuitively, it enforces a style constructed from an interpolation of the $k$ style prototypes.

**Optimization.** In pixel level attacks, two kinds of optimization techniques are widely used: *Gradient Sign Method*, e.g., PGD (Madry et al., 2018), and using continuous function, e.g. tanh, to approximate and bound $\ell_\infty$, e.g., in C&W (Carlini & Wagner, 2017). However in our context, we found these two techniques do not perform well. Using gradient sign tends to induce a large content loss while using tanh function inside the feature space empirically causes numerical instability. Instead, we use the iterative gradient method with gradient clipping. Specifically, We first calculate the gradient of loss $\mathcal{L}$ with respect to variables (e.g., $\boldsymbol{\tau}_c^\mu$ and $\boldsymbol{\tau}_c^\sigma$). The gradient is then clipped by a constant related to the dimension of variables. $||\nabla\mathcal{L}||_\infty \le 10/\sqrt{\text{Dimension of variable}}$. Then an Adam optimizer iteratively optimizes the variables using the clipped gradients.

## 4 EVALUATION

We evaluate feature space attack on 3 widely used image classification datasets and 5 state-of-the-art detection/defense approaches. All the target models (under attack) used in the experiments are pre-trained or trained using the code provided by the detection/defense approaches. We discuss how the detection/defense approaches perform under $\ell$-norm based pixel space attack in comparison with our feature space attack. Details are elaborated in the remainder of the section.

### 4.1 SETUP

**Datasets.** Three datasets are employed in the experiments: CIFAR-10 (Krizhevsky et al., 2009), ImageNet (Russakovsky et al., 2015) and SVHN (Netzer et al., 2011). CIFAR-10 is an object classification dataset, which consists of 10 classes. ImageNet is one of the largest image classification datasets and comprises 1,000 categories of objects. SVHN is a real-world digital recognition dataset with 10 classes of digits from 0 to 9.

**Detection and Defense Approaches.** We use 5 state-of-the-art detection and defense approaches to demonstrate the effectiveness of proposed feature space attack. Detection approaches aim to identify adversarial samples while they are provided to a DNN. They often work as an add-on to the model and do not aim to harden the model. We use a state-of-the-art adversarial example detection approach proposed by Roth et al. (2019) to test our attack. Defense approaches, on the other hand, harden models such that they are robust against adversarial example attacks. Existing state-of-the-art defense mechanisms either use adversarial training or certify a bound for each input image. We adopt 4 state-of-the-art defense approaches in the literature (Madry et al., 2018; Zhang et al., 2019; Xie et al., 2019; Song et al., 2019) for evaluation.

**Attack Settings.** The two proposed feature space attacks have similar performance on various experimental settings. Unless otherwise stated, we use feature augmentation attack as the default method. For the Encoder, we use VGG-19 from the input layer up to the relu4_1 for ImageNet, and up to relu2_1 for CIFAR-10 and SVHN . To launch attacks, we set the $\ell_\infty$-norm of embedding, $\epsilon$, in Equation 9 to $\ln(1.5)$ for all the untargeted attacks and $\ln(2)$ for all the targeted attacks. We randomly select 1,000 images to perform the attacks on ImageNet. For CIFAR-10 and SVHN, we use all the inputs in the validation set.

### 4.2 ATTACK AGAINST DETECTION APPROACH

We use a state-of-the-art adversarial sample detection approach "The Odds are Odd" (O2) (Roth et al., 2019) to demonstrate how feature space attack can evade it. O2 detects adversarial samples by adding random noise to input images and observing activation changing at a certain layer of a DNN. Specifically, O2 uses the penultimate layer (before the logits layer) as the representation of input images. It then defines a statistical variable that measures pairwise differences between two classes computed from the penultimate layer. The authors observed that adversarial samples differ significantly from benign samples regarding this variable when random noise is added. By performing statistical test on this variable, O2 is able to detect PGD attacks (Madry et al., 2018) with over 99% detection rate on CIFAR-10 with bound $\ell_\infty = 8/255$ and on ImageNet with $\ell_\infty = 2/255$. It also has over 90% detection rate against PGD and C&W (Carlini & Wagner, 2017) attacks under $\ell_2$ metric on CIFAR-10. The $\ell_2$ bounds were not given in the original paper (Roth et al., 2019).

Table 1 shows the results of O2 on detecting different input samples. The first two columns are the datasets and models used for evaluation. The third column denotes the prediction accuracy of models on normal inputs. The following three columns present the detection rate of O2 on normal inputs, PGD adversarial samples, and feature space adversarial samples, respectively. The detection rate on normal inputs indicates that O2 falsely recognizes normal inputs as adversarial, which are essentially false positives. We can observe that O2 can effectively detect PGD attack on both datasets, but fails to detect feature space attack. Particularly, O2 has only 0.04% detection rate on CIFAR-10, which indicates that *O2 is almost completely evaded by feature space attack*. As for ImageNet, O2 can detect 25.30% of feature space adversarial samples but at the cost of a 19.20% false positive rate[1]. The results show that O2 is ineffective against feature space attack.

---

[1]The parameters used for ImageNet are not given in the original paper. We can only reduce to this false positive rate after parameter tuning.

Table 1: O2 detection rate on normal inputs and adversarial samples.

| Dataset | Model | Accuracy | Detection Rate | | |
|---|---|---|---|---|---|
| | | | Normal | PGD | Feature Space |
| CIFAR-10 | ResNet-18 | 91.95 | 0.95 | 99.61 | **0.04** |
| ImageNet | ResNet-50 | 75.20 | 19.20 | 99.40 | **25.30** |

Table 2: Evaluation of adversarial attacks against various defense approaches.

| Attack | SVHN | CIFAR-10 | | ImageNet | | |
|---|---|---|---|---|---|---|
| | Adaption | Madry | TRADES | Denoise (t,1) | Denoise (u,1) | Denoise (u,5) |
| None | 84.84 | 77.84 | 84.97 | 61.25 | 61.25 | 78.12 |
| PGD | 52.84 | 41.43 | 54.02 | 42.60 | 12.50 | 27.15 |
| Decoder | 84.81 | 77.35 | 84.01 | 64.68 | 64.00 | 82.37 |
| Feature Space | **2.56** | **7.05** | **8.64** | **11.41** | **1.25** | **1.25** |

### 4.3 ATTACK AGAINST DEFENSE APPROACHES

We evaluate our feature space attack on 4 state-of-the-art adversarial training approaches: Madry (Madry et al., 2018), TRADES (Zhang et al., 2019), Denoise (Xie et al., 2019), and Adaption (Song et al., 2019). For Denoise, the original paper only evaluated on targeted attacks. We conduct experiments on both targeted and untargeted attacks. We hence use Denoise (t,1) to denote the top-1 accuracy of hardened model on targeted attack and Denoise (u,5) the top-5 accuracy on untargeted attack. We launch the PGD $\ell_\infty$ attack as well as our feature space attack on the four defense approaches. The experimental results demonstrate the performance of proposed feature space attack compared to existing $\ell$-norm based pixel space attack. Table 2 demonstrates the performance of adversarial attacks against various defense approaches. The first column denotes attack methods, where "None" presents the model accuracy on benign inputs and "Decoder" denotes the samples directly generated from the decoder without any feature space perturbation. The latter is to show that the Decoder can generate faithful and natural images from embeddings. The following columns show different defense approaches (second row) applied on various datasets (first row). We can see that the PGD attack can reduce model accuracy to some extent when defense mechanisms are considered. *Feature space attack, on the other hand, can effectively reduce model accuracy down to less than 12%, and most results are one order of magnitude smaller than PGD.* Especially, model accuracy on ImageNet is only 1.25% when using untargeted attack, even in the presence of the defense technique. Interestingly, if images are generated directly from the Decoder without any feature space perturbation, the model accuracy improves on ImageNet. This indicates that the trained Decoder indeed captures the content feature of input images.

We study the $\ell$-norm distances in both the pixel space and the feature space for both pixel space attacks and feature space attacks. We observe that in the pixel space, the introduced perturbation by feature space attack is much larger than that of the PGD attack. However in the feature space, our attack has very similar distances as PGD. Figure 1 and Figure 3 (in Appendix §A.1) show that the adversarial samples have only style differences that are natural or even human imperceptible. Details can be found in Appendix §A.1.

We conducted an experiment to show that adversarial training in the pixel space does not improve model robustness against feature space attack and vice versa. Details can be found in Appendix §A.1. We have also studied the characteristics of the adversarial samples generated by different feature space attacks and attack settings. Please see Appendix §A.2.

## 5 CONCLUSIONS

We propose feature space adversarial attack on DNNs. It is based on perturbing style features and retaining content features. Such attacks inject natural style changes to input images to cause model misclassification. Since they usually cause substantial pixel space perturbations and existing detection/defense techniques are mostly for bounded pixel space attacks, these techniques are not effective for feature space attacks.

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

# A    APPENDIX

## A.1    MEASUREMENT OF PERTURBATION IN PIXEL AND FEATURE SPACES

To measure the magnitude of perturbation introduced by adversarial attacks, we use both $\ell_\infty$ and $\ell_2$ distances. In addition, as we aim to understand how the different attacks perturb the pixel space and the feature space, we compute the distances for both spaces. For the pixel space, the calculation is discussed in §2. For the feature space, we normalize the embeddings before distance calculation. For each channel, we use $h(\boldsymbol{x}) = \frac{f(\boldsymbol{x}) - \mu_{f(\boldsymbol{x})}}{\sigma_{f(\boldsymbol{x})}}$ to normalize the embedding produced by the Encoder $f(\cdot)$ given input $\boldsymbol{x}$. The feature space difference hence can be computed as $||h(\boldsymbol{x}) - h(\boldsymbol{x}')||_p$. Table 3, Table 5 and Table 6 illustrate the magnitude of perturbation introduced by adversarial attacks on pixel and feature spaces for different hardened models. It can be observed that in the pixel space, the introduced perturbation by feature space attack is much larger than that of the PGD attack with $\ell_\infty$ and $\ell_2$ distances. In the feature space, however, our feature space attack does not induce large difference between normal inputs and adversarial samples. Particularly, the difference is similar or even smaller than that by the PGD attack. We further investigate the feature space distance between feature space adversarial samples and the corresponding normal images of their target labels. The $\ell_\infty$ and $\ell_2$ distances are 24.24 and 890 on the hardened (targeted) ImageNet model respectively, which are much larger than those (9.99 and 283) between the adversarial samples and the original (attacked) images. This indicates that feature space attack indeed leverages abstract features for generating adversarial samples. As shown in Figure 1 and Figure 3, the introduced perturbation is either insensitive to humans or even imperceptible.

We further explore the space targeted by different adversarial attacks using adversarial training. The assumption is that models hardened using an attack method are resilient to adversarial samples generated by the same attack method. Specifically, we use a CIFAR-10 model hardened by PGD as the base model. We then employ different attack approaches to generate adversarial samples against this base model on both the training and validation sets. For each attack approach, we further retrain the base model using the corresponding adversarial samples. Finally, we test the performance of the retrained model on different adversarial samples from the validation set. Table 4 presents the results. Each row represents the model retrained using the corresponding adversarial samples on the training set, where FA is feature augmentation attack and FI feature interpolation attack. Each column denotes the test on adversarial samples generated on the validation set. We observe that retraining on the PGD attack does not improve model robustness against feature space attacks and vice versa. It indicates that the PGD attack and feature space attack exploit different spaces in generating adversarial samples.

Table 3: Magnitude of perturbation on hardened SVHN models.

| Attack | Pixel Space | | Feature Space | |
|---|---|---|---|---|
| | $l_\infty$ | $l_2$ | $l_\infty$ | $l_2$ |
| PGD | 0.02 | 1.04 | 11.50 | 53.78 |
| Decoder | 0.08 | 1.05 | 10.12 | 36.52 |
| Feature Space | 0.12 | 2.01 | 10.51 | 41.83 |

Table 4: Model accuracy of adversarially trained models in different attack spaces.

| Model | PGD | FA | FI |
|---|---|---|---|
| PGD | 72.18 | 55.31 | 57.96 |
| FA | 54.68 | 84.06 | 75.31 |
| FI | 46.56 | 60.93 | 87.81 |

Table 5: Magnitude of perturbation on hardened CIFAR-10 models.

| Attack | Madry | | | | TRADES | | | |
|---|---|---|---|---|---|---|---|---|
| | Pixle Space | | Feature Space | | Pixle Space | | Feature Space | |
| | $\ell_\infty$ | $\ell_2$ | $\ell_\infty$ | $\ell_2$ | $\ell_\infty$ | $\ell_2$ | $\ell_\infty$ | $\ell_2$ |
| PGD | 0.03 | 1.51 | 6.86 | 41.71 | 0.03 | 1.47 | 6.49 | 37.89 |
| Decoder | 0.19 | 1.85 | 4.29 | 25.13 | 0.18 | 1.85 | 4.30 | 25.15 |
| Feature Space | 0.27 | 4.08 | 6.88 | 43.38 | 0.28 | 4.72 | 7.43 | 46.43 |

Table 6: Magnitude of perturbation on hardened ImageNet models.

| Attack | Targeted | | | | Untargeted | | | |
|---|---|---|---|---|---|---|---|---|
| | Pixle Space | | Feature Space | | Pixle Space | | Feature Space | |
| | $\ell_\infty$ | $\ell_2$ | $\ell_\infty$ | $\ell_2$ | $\ell_\infty$ | $\ell_2$ | $\ell_\infty$ | $\ell_2$ |
| PGD | 0.06 | 19.48 | 12.08 | 375 | 0.03 | 9.24 | 16.91 | 227 |
| Decoder | 0.87 | 44.98 | 8.26 | 193 | 0.88 | 44.67 | 15.24 | 214 |
| Feature Space | 0.89 | 69.07 | 9.99 | 283 | 0.86 | 54.89 | 16.09 | 236 |

## A.2 TWO FEATURE SPACE ATTACKS

We generate and visually analyze the two feature space attacks. The ImageNet model hardened by feature denoising is used for generating adversarial samples. Columns (a), (d), and (g) in Figure 3 present the original images. Columns (b), (c), and (d) present the adversarial samples generate by the Encoder and the corresponding Decoder, with different encoder depths. Specifically, column (b) uses conv2_1 layers, column (c) uses conv3_1 layers and column (d) uses conv4_1 layers of a pre-trained VGG-19 as the Encoder, and the Decoders are of neural network structure similar to the corresponding Encoders but in a reverse order. Observe that as the Encoder becomes deeper, object outlines and textures are changed in addition to colors. Columns (e) and (f) are adversarial samples

generated by the feature argumentation attack (FA) and feature interpolation attack (FI). We observe that they both generate realistic images.

In column (h), we only perturb the mean of embedding whereas in column (i) we only perturb the standard deviation of embedding. The results indicate that mean values tend to represent the background and the overall color tone. In contrast, the standard deviations tend to represent the object shape and relative color.

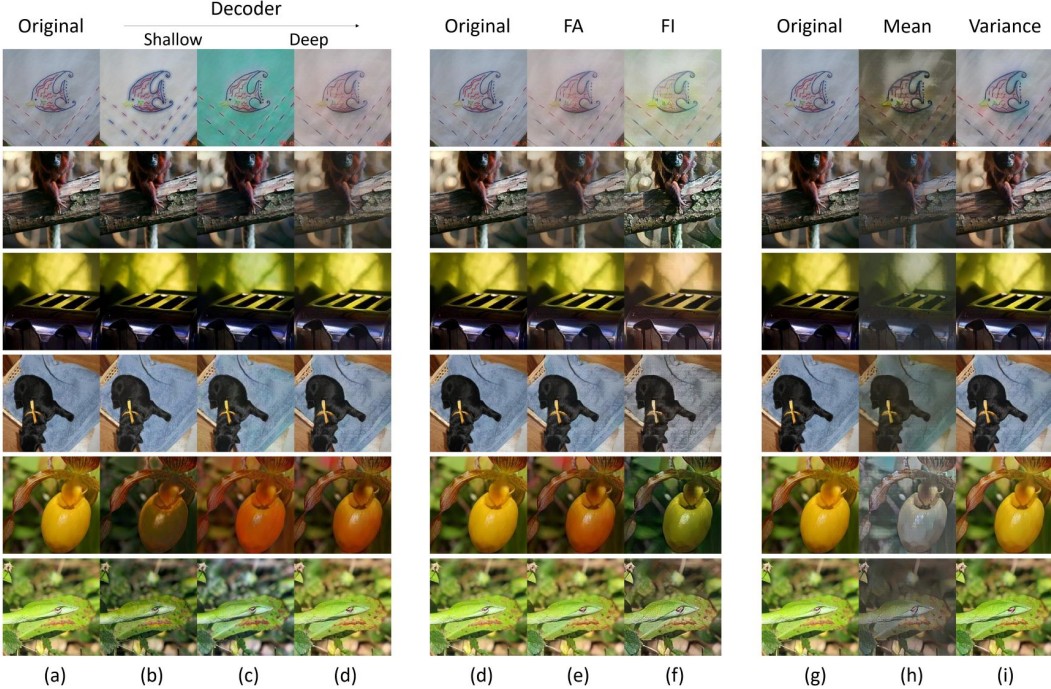

Figure 3: The adversarial samples from different feature space attack methods.

