# OpenReview forum: "TOWARDS FEATURE SPACE ADVERSARIAL ATTACK"
_ICLR.cc/2020/Conference — Reject_

### Official Review · AnonReviewer1 · 2019-10-22
**Official Blind Review #1**

**Rating:** 6

**Review:**

This paper presents an interesting new adversarial attack technique that attempts to perturb abstract features learned by the target neural net. It is well written and easy to follow.  Its main contribution is the joining of ideas developed in the style transfer literature with those from the adversarial literature.

The authors establish that they are able to create adversarial attacks that look similar to the original image but are miss classified. These images are not bounded by small epsilons, but are said to be indistinguishable by people. They illustrate a sample of these attacks, but no human study is employed to back up this claim. A simple human evaluation to prove that the attacks are indistinguishable from unperturbed images would strengthen the work (This can be done easily and at low cost by  employing mechanical Turk or an equivalent system for example).

They make use of a detection mechanism (The Odds are Odd, Rothet al.) to verify that their adversarial attacks are hard to detect, but this particular detection mechanism has already  been broken (https://arxiv.org/pdf/1907.12138.pdf). If there is an as of yet unbroken detection mechanism that could be tested, that would improve the work. Alternatively the authors should acknowledge that there are simpler ways of evading this detection method.

The attack that they propose targets the feature space, but no feature space detection methods are tested. The work would be improved by testing on a feature based detection methods such as dKNN (https://arxiv.org/pdf/1902.01889.pdf)

Overall the work is interesting and novel, and creatively joins together two otherwise distinct areas of machine learning research to make a modest but novel contribution to the field.

**Experience Assessment:**

I have published one or two papers in this area.

**Review Assessment: Checking Correctness Of Derivations And Theory:**

N/A

**Review Assessment: Checking Correctness Of Experiments:**

I assessed the sensibility of the experiments.

**Review Assessment: Thoroughness In Paper Reading:**

I read the paper at least twice and used my best judgement in assessing the paper.

---

> ### Author Response · Authors · 2019-11-13
> **Response to Review #1**
>
> Thanks for your appreciation and constructive comments, we list your concerns and answer them as follows.
>
> Review #1
> R1Q1: human study
>
> We add a user study on AMT to compare the perceptual quality of feature space attack with PGD.
> We randomly selected 200 pairs of adversarial samples, each consisting of an adversarial example from PGD and the other from our feature space attack. Every pair is repeated for 3 times, making a total of 600 pairs. The adversarial samples are for targeted attack on ImageNet, where PGD has 58% successful rate and Feature Space Attack has 88% successful rate.
> The worker is asked “Which image appears more natural, reasonable and realistic to you? Choose left or right to indicate your choice.” The order in pair is shuffled.
> 40% of users chose samples Feature Space Attack and 60% users chose samples from PGD. It shows that although the images by PGD are slightly more natural than ours. They are comparable.
> Also, we provide a set of adversarial samples for inspection along with the code at https://github.com/JerishDansolBalala/FeatureSpaceAtk .
>
> R1Q2: unbroken detection mechanism
> We were not aware of that “The Odds are Odd” has been broken during the submission period and will add this information in the paper. The attack [4] specifically designed for the detection approach requires a much stronger threat model, where the attacker already knows the existence of the defense. In our case, however, we are able to evade the detection method without knowing its existence or mechanism.
>
> R1Q3: add dKNN as feature space detection
> We use the same setting as in the original paper and the code provided by authors to conduct the experiments. We test on CIFAR-10  and two models (CNN+MLP in the default setting and ResNet18). For CNN+MLP, the detection rate against PGD attack is 3.9% and our feature space attack 1.9%. For ResNet-18, the detection rate against PGD attack is 11% and ours 5.4%. Our proposed feature space attack is stronger than pixel-level attacks, and can effectively evade feature based detection methods.
>
> [1] H. Hosseini, R. Poovendran. Semantic Adversarial Examples. https://arxiv.org/abs/1804.00499
> [2] C. Laidlaw, S. Feizi. Functional Adversarial Attacks. https://arxiv.org/pdf/1906.00001
> [3] Y. Song, R. Shu, N. Kushman, S. Ermon. Constructing unrestricted adversarial examples with generative models. https://arxiv.org/abs/1805.07894
> [4] H. Hosseini, S. Kannan, R. Poovendran, Are Odds Really Odd? Bypassing Statistical Detection of Adversarial Examples. https://arxiv.org/pdf/1907.12138.pdf
> [5] L. A. Gatys, A. S. Ecker, and M. Bethge. Image style transfer using convolutional neural networks. In CVPR, 2016.
> [6] D. Ulyanov, A. Vedaldi, V. Lempitsky. Instance Normalization: The Missing Ingredient for Fast Stylization. https://arxiv.org/abs/1607.08022
> [7] X. Huang and S. Belongie. Arbitrary style transfer in real-time with adaptive instance normalization. In ICCV, 2017

---

### Official Review · AnonReviewer3 · 2019-10-23
**Official Blind Review #3**

**Rating:** 6

**Review:**

This paper presents an adversarial attack method, which conducts perturbations in the feature spaces, instead of the raw image space. Specifically, the proposed method firstly learns an encoder that encodes features into the latent space, where style features are learned. At the same time, a decoder is learned to reconstruct the images with the encoded features. To conduct attacks, perturbations are added into the encoded features and attack images are generated with the decoder given the perturbated features. The experiment results look promising, showing that the proposed method achieves better attack performance with realistic adversarial images.

The general idea of perturbating the feature (latent) space is not a novel one, which has been studied in [1]. However, the proposed one is with an autoencoder framework instead of GAN used in [1]. Therefore, the proposed approach is able to construct adversarial examples for specific images. In addition, the training of the encoder is adapted from a style transfer method, which seems to learn good features that capture style features.

It is a bit unclear on the intuition of the constructions of Eq. (5) and (6). The details may be in Huang & Belongie, 2017. But it is better to provide more intuitive explanation and discussion on why these constructions capture style variation.

The results shown in the paper look promising. But it would be more comprehensive to compare with other pixel attacks in addition to PGD. Moreover, it is unclear whether it is a fair comparison between the proposed approach and pixel attacks, even under the same amount of perturbations. It would be good if the code will be released.

Minor:

Last sentence in the first paragraph of page 3: a missing reference.

[1] Song, Yang, Rui Shu, Nate Kushman, and Stefano Ermon. "Constructing unrestricted adversarial examples with generative models." In Advances in Neural Information Processing Systems, pp. 8312-8323. 2018.

**Experience Assessment:**

I have read many papers in this area.

**Review Assessment: Checking Correctness Of Derivations And Theory:**

I assessed the sensibility of the derivations and theory.

**Review Assessment: Checking Correctness Of Experiments:**

I assessed the sensibility of the experiments.

**Review Assessment: Thoroughness In Paper Reading:**

I read the paper at least twice and used my best judgement in assessing the paper.

---

> ### Author Response · Authors · 2019-11-13
> **Response to Review #3**
>
> Thanks for your constructive comments and appreciation. Here we list our response as follows.
>
> Review #3
> R3Q1: intuition of Eq. (5) and (6)
> The choice of mean and variance as style is largely an empirical observation. Previous work observed that normalized output of shallow convolutional layers keeps the shape of images. Paper [5] found statistics (e.g., correlation of feature maps) of an image represent style. Paper [6] found that instance normalization (IN) increases the quality of style transfer. Paper [7] investigated IN and proposed to use AdaIN, which transfer mean and variance only at a given layer. These experiment results from previous work show that statistics from model layers carry style information. We will add the discussion to the paper.
>
> R3Q2: Compare to other pixel space attacks and code release
> We additionally compared to pixel attacks FGSM, CW and DeepFool on CIFAR10. They have the same bound as PGD. The accuracies are respectively 61.06% for FGSM, 61.38% for DeepFool, 81.24% for CW. Compared with 8.64% for feature space attack and 54.02% for PGD. Our comparison aims to show the different nature of pixel attacks and feature attacks renders existing pixel defense ineffective. We have additionally experimented a recent feature space defense. Please see R1Q3 and R2Q2.
>
> We release the code at https://github.com/JerishDansolBalala/FeatureSpaceAtk .
>
> R3Q3: Missing citation:
> Paper [3] is a closely related work. It adds additional losses to GAN for generating unrestricted adversarial samples.  A vanilla GAN model only generates over a distribution of limited support, while an autoencoder can attack specific sample. We will add this discussion to the paper.
>
>
> [1] H. Hosseini, R. Poovendran. Semantic Adversarial Examples. https://arxiv.org/abs/1804.00499
> [2] C. Laidlaw, S. Feizi. Functional Adversarial Attacks. https://arxiv.org/pdf/1906.00001
> [3] Y. Song, R. Shu, N. Kushman, S. Ermon. Constructing unrestricted adversarial examples with generative models. https://arxiv.org/abs/1805.07894
> [4] H. Hosseini, S. Kannan, R. Poovendran, Are Odds Really Odd? Bypassing Statistical Detection of Adversarial Examples. https://arxiv.org/pdf/1907.12138.pdf
> [5] L. A. Gatys, A. S. Ecker, and M. Bethge. Image style transfer using convolutional neural networks. In CVPR, 2016.
> [6] D. Ulyanov, A. Vedaldi, V. Lempitsky. Instance Normalization: The Missing Ingredient for Fast Stylization. https://arxiv.org/abs/1607.08022
> [7] X. Huang and S. Belongie. Arbitrary style transfer in real-time with adaptive instance normalization. In ICCV, 2017

---

### Official Review · AnonReviewer2 · 2019-10-23
**Official Blind Review #2**

**Rating:** 3

**Review:**

Authors have introduced a new type of adversarial attacks that perturb abstract features of the image. They have shown that pixel space adversarial attack detection and defense techniques are ineffective in guarding against feature space attacks.

I have some concerns about the novelty of the attack and the appropriateness of defenses that have been tested.

- Since the attack is done in the feature space, the defense should also be done in the feature space. For example, adversarial training or smoothing can be done in the feature space. See: https://arxiv.org/abs/1802.03471

- There are attacks that perturb colors or other interpretable features of the image that have not been mentioned in the paper. For example, see https://arxiv.org/abs/1804.00499 and https://arxiv.org/pdf/1906.00001

- If the decoder has a high Lipschitz constant, a small perturbation in the feature scape 'can' lead to a large and visible perturbation in the pixel space. It was not clear to me how this is being controlled in the current method.


**Experience Assessment:**

I have published in this field for several years.

**Review Assessment: Checking Correctness Of Derivations And Theory:**

I did not assess the derivations or theory.

**Review Assessment: Checking Correctness Of Experiments:**

I assessed the sensibility of the experiments.

**Review Assessment: Thoroughness In Paper Reading:**

I made a quick assessment of this paper.

---

> ### Author Response · Authors · 2019-11-13
> **Response to Review #2**
>
> We thank for your constructive comments. Here we list your concerns and answer them one by one.
>
> R2Q1: novelty
> It is a novel way to conduct feature space attack with the combination of style transfer and manipulation of model internal embedding, as pointed out by Review #1. Common style transfer task requires additional information such as painting styles. In this paper, we instead use samples from the same class that share rich style features, which is not explored by existing work. We leverage these implicitly learned features to launch our feature-space attack, which distinguishes ours among various existing attack methods.
> Unlike in [3], where a vanilla GAN-based attack method generates over a distribution of limited support, and has no control of the generated samples, our encoder-decoder based structure enables attacking each individual sample with controlled content and there is no limit on the number of samples, which is also mentioned in Review #3.
> We also conducted an experiment that explores the defensive methods in different spaces. We observed that pixel-level defense is not effective against feature-space attack, and as we will show in the R2Q2 and R1Q3, existing defense that can be used in the feature space cannot defend our attack.
>
> R2Q2: add Pixel-DP as possible feature space defense
> We use the code provided by authors and the same setting to conduct experiments on Pixel-DP defense. When using l2 norm bound of 0.1, the model accuracy (with Pixel-DP defense) is 80% under PGD l2 attack and 0% under our feature space attacks. When using l2 norm bound of 1, the model accuracy (with Pixel-DP defense) is 31% under PGD l2 attack and 0% under ours. When further increasing l2 norm bound to 10, we found the accuracy on normal images degrades to below 15%. Pixel-DP is hence ineffective against our feature space attack. The results are reasonable as Pixel-DP can only certify the l_2 norm bound up to 1, whereas our feature space attack generates adversarial samples with l_2 norm usually larger than 10.
> We have also conducted adversarial training experiments over pixel/feature spaces, which is shown in Table 4 in appendix. When the model is trained with one type of adversarial attacks, it can only achieve non-trivial accuracy on the same attack but is ineffective to other types of attacks. It is non-trivial to design an effective adversarial training that is robust to all types of adversarial attacks, and it is out of scope of this paper.
>
> R2Q3: color space attacks
> The paper [1] proposed to modify the HSV color space to generate adversarial samples, which transforms all pixels by a non-parametric function uniformly.  The experiments were only conducted on CIFAR-10 dataset and Madry model. Differently, feature space attack changes colors of specific objects or background alone and the transformation is learned from similar images, which is more imperceptible.  The paper [2] proposed to change the lighting condition and color similarly in [1] to generate adversarial samples. We observe in experiments that feature space attack also learns to modify lightning condition, color and texture as well, please refer to Fig. 3. Compared to these, our attack is more general, and the features attacked are more subtle. We will include the discussion in the paper.
>
> R2Q4: large and visible perturbation
> Our Decoder is trained to minimize difference between the reconstructed image and the original image. We only modify the mean and variance of activations that are considered style of features [5], where the content of images is preserved. When launching attack, we bound the perturbation of mean and variance, which controls the perturbation introduced in the pixel space. Fig. 3, Fig. 5 and Fig. 6 show the pixel/feature space distance of all the generated adversarial samples. It can be observed that our feature space attack has small feature space distance and even smaller than (pixel-space l_inf bounded) PGD attack in most cases. And we conducted an additional user study, suggesting feature space attack has comparable perceptual quality as PGD, please refer to R1Q1.
>
> [1] H. Hosseini, R. Poovendran. Semantic Adversarial Examples. https://arxiv.org/abs/1804.00499
> [2] C. Laidlaw, S. Feizi. Functional Adversarial Attacks. https://arxiv.org/pdf/1906.00001
> [3] Y. Song, R. Shu, N. Kushman, S. Ermon. Constructing unrestricted adversarial examples with generative models. https://arxiv.org/abs/1805.07894
> [4] H. Hosseini, S. Kannan, R. Poovendran, Are Odds Really Odd? Bypassing Statistical Detection of Adversarial Examples. https://arxiv.org/pdf/1907.12138.pdf
> [5] L. A. Gatys, A. S. Ecker, and M. Bethge. Image style transfer using convolutional neural networks. In CVPR, 2016.
> [6] D. Ulyanov, A. Vedaldi, V. Lempitsky. Instance Normalization: The Missing Ingredient for Fast Stylization. https://arxiv.org/abs/1607.08022
> [7] X. Huang and S. Belongie. Arbitrary style transfer in real-time with adaptive instance normalization.

---

### Decision · Program_Chairs · 2019-12-19

**Decision:**

Reject

**Comment:**

This paper provides an improved feature space adversarial attack.

However, the contribution is unclear in its significance, in part due to an important prior reference was omitted (song et al.)

Unfortunately the paper is borderline, and not above the bar for acceptance in the current pool.